# Experimental Results of Three-Dimensional Modeling and Mapping with Airborne Ka-Band Fixed-Baseline InSAR in Typical Topographies of China

Jian Gao [1], Zhongchang Sun [2,3,*], Huadong Guo [3], Lideng Wei [4], Yongjie Li [5] and Qiang Xing [3]

1    School of Geographic and Biologic Information, Nanjing University of Posts and Telecommunications, Nanjing 210023, China; gaoj@njupt.edu.cn
2    Key Laboratory of Earth Observation Hainan Province, Sanya Institute of Remote Sensing, Sanya 572029, China
3    Key Laboratory of Digital Earth Science, Aerospace Information Research Institute, Chinese Academy of Sciences, Beijing 100094, China; hdguo@radi.ac.cn (H.G.); xingqiang@aircas.ac.cn (Q.X.)
4    Beijing Institute of Radio Measurement, Beijing 100854, China; weilideng@tsinghua.org.cn
5    China Coal Zhongyuan (Tianjin) Construction Supervision Consulting Co., Ltd., Tianjin 300120, China; liyongjie1256@163.com
*    Correspondence: sunzc@aircas.ac.cn; Tel.: +86-010-82178093

**Abstract:** Interferometric synthetic aperture radar (InSAR) has become a key technology for producing high-precision digital surface models (DSMs) and digital orthophoto maps (DOMs) in full time and all weathers. Airborne millimeter-wave InSAR, with large-scale and high-resolution imaging, is characterized by high spatial resolution, flexibility, and immunity to loss-of-correlation. This paper introduces our modeling experiments with airborne dual-antenna, Ka-band InSAR regarding typical topographies of China. Ka-band SAR data were acquired in designated experimental areas in flat (Heyang area in Shaanxi) and mountainous areas (Shibing area in Guizhou and Qionglai area in Sichuan). The key processing of the experimental data for DSMs and DOMs is demonstrated in the paper, especially the proposed robust and efficient phase unwrapping (PU) method for the interferometric data and block adjustment method of strip calibration. The results show that the proposed unwrapping method can provide reliable unwrapped phase results in undulating areas, and the block adjustment can carry out consistent calibration for strips with sparse ground control points (GCPs). The accuracy assessment of the DSM shows that the coordinate root mean square error (RSME) of the obtained DSM is less than 2 m in height, and 2.5 m horizontally, which meets the 1:5000 requirement for topographic mapping in difficult areas.

**Keywords:** three-dimensional modeling; topographic mapping; airborne millimeter-wave; Ka-band; DOM; DSM

## 1. Introduction

Traditional field surveying and mapping are time-consuming and labor-intensive, and unsuitable for obtaining large-scale digital surface models (DSMs) and digital orthophoto maps (DOMs). Although Lidar technology can help acquire DSMs precisely, its technical threshold is relatively high. When a DSM is acquired using satellite remote sensing and photogrammetry, it is often affected by clouds, fog, rain, and other weather conditions. The quality and accuracy of data might deteriorate, possibly resulting from a complicated geographic environment and the vagaries of climate. Interferometric synthetic aperture radar (InSAR) can obtain data in full time and weather on a large scale, overcoming the influence of severe weather conditions [1,2]. The popular spaceborne InSAR can cover a wide area, and its satellite orbit is relatively fixed. However, the spaceborne observation area cannot be alternated for individual tasks, and the repeat-pass nature leads to unpredictable temporal correlation loss and atmospheric influence [3]. Airborne

platforms have better availability and adaptability for local SAR observations. Since the 1980s, dual-antenna airborne InSAR has been an effective method for three-dimensional modeling in local areas, generating DSM and DOM products [4]. The resolution of the experimental results has been improved from the early ten-meter level [5–8] to the submeter level [9], and the elevation accuracy can also reach the submeter level [10].

The millimeter-wave radar system takes the Ka-band electromagnetic signal, whose frequency is higher and closer to that of visible light, compared with the C, S, X, and Ku bands. The available frequency bandwidth of the millimeter-wave system is wider, and its smaller diameter antenna can obtain a narrower antenna beam and higher antenna gain. Thus, high-resolution SAR data are achievable in both the range and azimuth directions. The shorter wavelength and weak penetration are more conducive to DSM acquisition [11] in InSAR. In addition, the size of the antenna and microwave components of a millimeter-wave system is correspondingly reduced, which is more conducive to realizing the miniaturization of the radar system and can be applied to unmanned aerial vehicles (UAVs). Millimeter-wave InSAR has been used for snow depth mapping [12], traffic obstacle detection [13], and even indoor navigation [14]. The airborne dual-antenna, millimeter-wave InSAR mode has high resolution, mobility, and flexibility [15,16], which allow for further application in topographic modeling.

The application of airborne InSAR technology faces some challenges. The airborne platform is susceptible to atmospheric turbulence during operation, and there is an unpredictable deviation between the real motion trajectory and the designed trajectory. The moving deviation could degrade the quality of SAR focusing. The position and orientation system (POS) can provide accurate dynamic position and orientation data, which play an important part in motion compensation information for SAR focusing [15,17]. The stability of the system parameters of the airborne platform is low during data collection. Terrain modeling requires precise parameters, and independent calibration processing is necessary. The closer the observation distance, the less the height ambiguity and higher resolution are likely to locally lower spatial coherence in undulating areas with shadow and layover compared with the spaceborne platform, which needs a more robust phase unwrapping (PU) processing method. In addition, the data acquired in scenes and strips in the target area should be used in a uniform coordinated system, which requires additional processes to achieve data registration and strip calibration.

In this paper, we introduce our topographic modeling work on Ka-band SAR data acquired at three experimental sites in typical topographies of China, including the flat Heyang area in Shaanxi Province, the mountainous Shibing area in Guizhou Province, and the Qionglai area in Sichuan Province. The special data-processing steps designed for the Ka-band InSAR procedure include (1) a reliable PU method using the minimum balanced trees (MBT) unwrapping algorithm in parallel-processing mode; (2) system parameter calibration and block adjustment with tie points (TPs) in overlapped strips for reconstructing a three-dimensional model; and (3) accuracy assessment of the provided modeling procedure for DOMs and DSMs, which is implemented in a self-developed InSAR data-processing tool, AirborneInSARMap.

## 2. Experimental Area and System Parameters

### 2.1. General Information of Experimental Areas

The three experimental sites include the Heyang, Shibing, and Qionglai areas. The Heyang experimental area is located in Heyang County, northwest of Weinan city, in the eastern part of Shaanxi Province. The longitude and latitude ranges of the experimental area are 110.061°E~110.096°E, 35.080°N~35.102°N and 3.19 km by 2.44 km. The altitude range is 546~705 m from the Shuttle Radar Topography Mission (SRTM) digital elevation model (DEM). The terrain of the experimental area is mainly plains, including towns and farmland. The Shibing experimental area is located in Shibing County in southeastern Guizhou Province. The longitude and latitude ranges of the experimental area are 108.184°E~108.185°E, 26.889°N~26.919°N and 14.99 km by 3.35 km. The altitude range

is 560~1198 m. The terrain of the experimental area is relatively rugged, with mountains, towns, roads, farmland, forests, etc. The Qionglai experimental area is located in Qionglai city in the center of Sichuan Province. The longitude and latitude ranges of the experimental area are 103.140°E~103.283°E, 30.319°N~30.493°N and 21.27 km by 2.56 km. The altitude range is 603~1344 m. The terrain of the experimental area is rather rugged and undulating, with high mountains, towns, and farmland. The three experimental sites are shown in Figure 1.

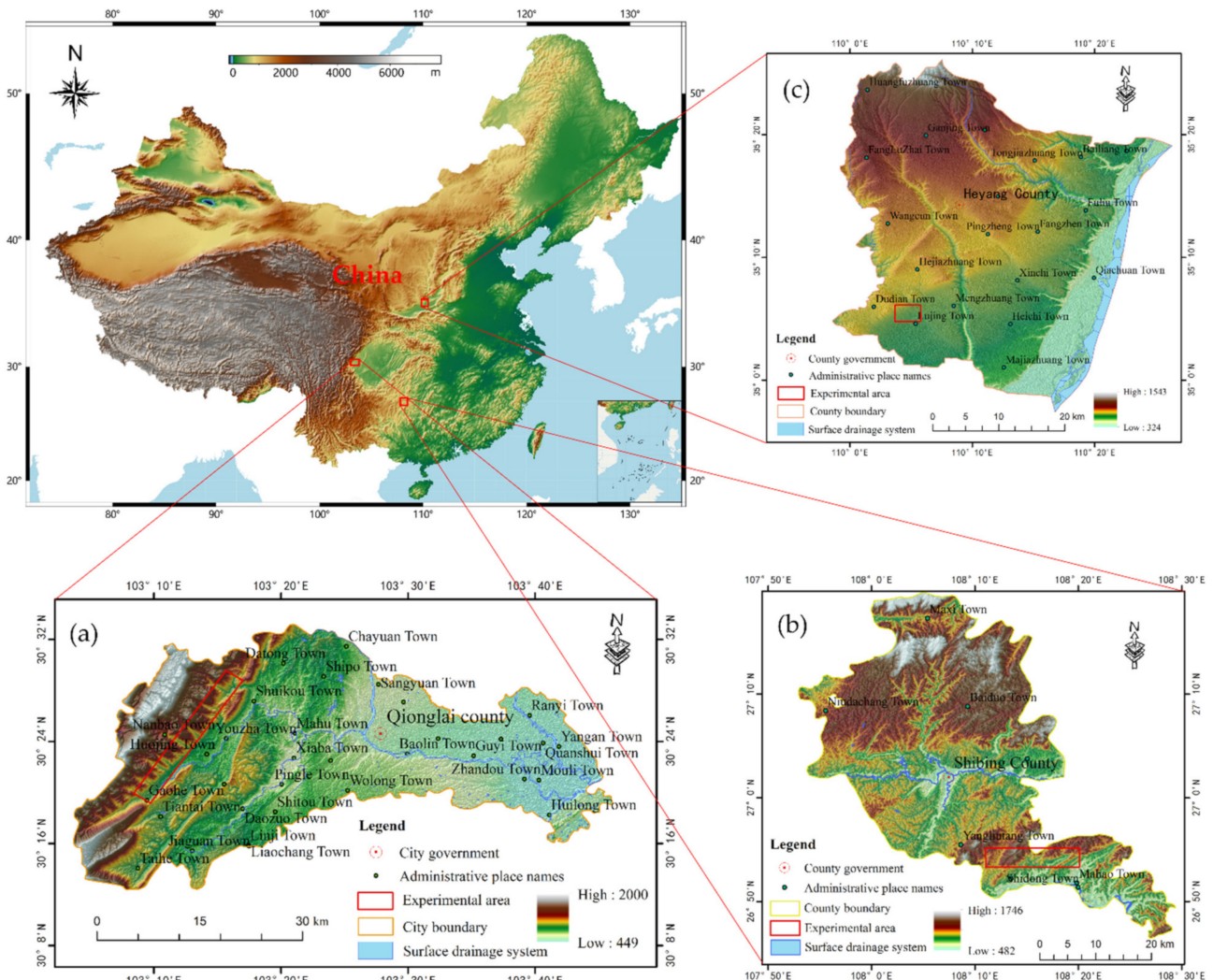

**Figure 1.** The three experimental areas for millimeter-wave InSAR data acquisition. (**a**) Qionglai area, (**b**) Shibing area, and (**c**) Heyang area.

## 2.2. Data Acquisition in the Experimental Areas

The observational equipment used in the experiment was an airborne Ka-band interferometric SAR system developed by the Beijing Institute of Radio Measurement, which includes multi-channel, multi-baseline, and multi-polarization, and is competent for aerial mapping, environmental monitoring, accurate disaster assessment, etc. Table 1 lists the airborne SAR system parameters. The carrier was equipped with a precise aviation POS that inherits the Global Navigation Satellite System (GNSS) and Inertial Navigation System (INS). During data acquisition, POS captured the position and orientation data hundreds of times per second, including three-dimensional coordinates, attitude angle, velocity, acceleration, etc.

**Table 1.** Experimental parameters of the airborne InSAR system.

| Parameter | Heyang | Shibing | Qionglai |
|---|---|---|---|
| Band | Ka | Ka | Ka |
| Wavelength/m | 0.008 | 0.008 | 0.008 |
| Frequency/GHz | 35 | 35 | 35 |
| Chirp bandwidth/MHz | 900 | 900 | 900 |
| Interferometric mode | 1 | 1 | 1 |
| Baseline/m | 0.313 | 0.313 | 0.313 |
| Azimuth row | 14,224 | 17,136 | 13,120 |
| Range column | 8192 | 8704 | 16,384 |
| Initial slant distance/m | 3599 | 3989 | 4618 |
| Carrier height/m | 3435 | 4043 | 4183 |
| Positioning accuracy (H)/m | 0.03 | 0.03 | 0.03 |
| Positioning accuracy (V)/m | 0.06 | 0.06 | 0.06 |
| Roll accuracy/deg | 0.0025 | 0.0025 | 0.0025 |
| Pitch accuracy/deg | 0.0025 | 0.0025 | 0.0025 |
| Azimuth resolution/m | 0.123 | 0.152 | 0.142 |
| Range resolution/m | 0.134 | 0.134 | 0.134 |

The airborne InSAR system was employed to obtain 4 scenes of single look complex (SLC) data (two strips) from west to east in the Heyang experimental area with 22 ground control points (GCPs, corner reflector, as shown in Figure 2) arranged at the beginning and end of the test area, including 9 GCPs for block adjustment and 13 GCPs for DSM verification. Twenty-one scenes of SLC data (three strips) were obtained from west to east in the Shibing experimental area with 26 GCPs, including 14 GCPs for block adjustment and 12 GCPs for DSM verification. There were 13 scenes of SLC data (one strip) obtained from southwest to northeast in the Qionglai experimental area with 25 GCPs, including 12 GCPs for block adjustment and 13 GCPs for DSM verification.

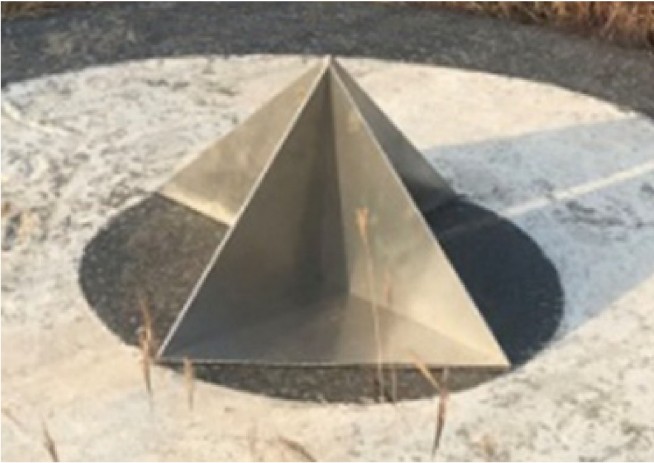

**Figure 2.** A metal corner reflector in the data acquisition experiments.

### 3. Three-Dimensional Reconstruction Model

The principle of InSAR is based on interferometry of two SLC data obtained in the observation area on two or more radar antennas with different viewing angles. According to the geometric relationship between the ground targets and antennas, complex images are differentiated to produce an interference phase map. During the interferometric phase, the slant distance difference between the antennas and targets on the ground can be obtained. The elevation of the ground targets can be calculated according to the known information, such as the speed, position, and sensor parameters of the observation platform, allowing the DSM of the observation area to be reconstructed [1].

Based on the orthogonal decomposition algorithm of the look vector [18,19], we converted the look vector from the Madsen moving coordinate (MMC) system to a ground coordinate system for three-dimensional reconstruction. A demonstration of the geometric relation in three-dimensional reconstruction is shown in Figure 3.

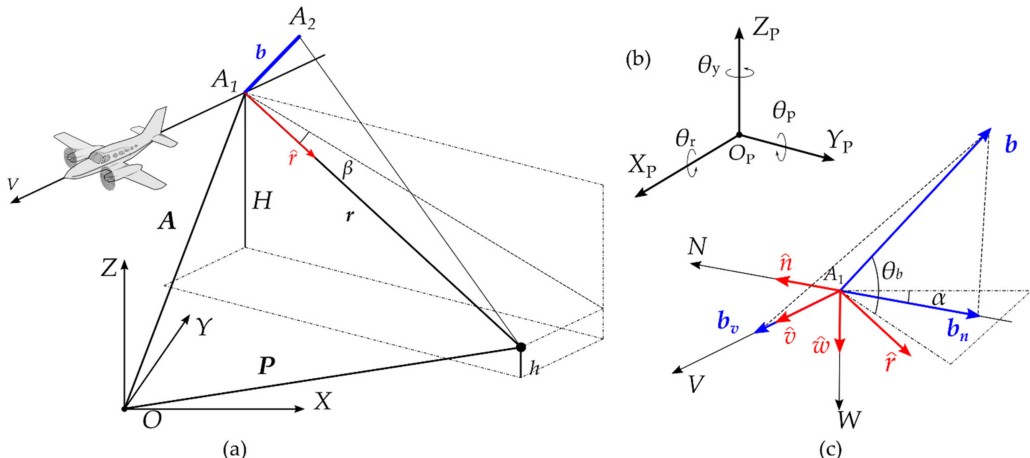

**Figure 3.** Three-dimensional geometry of airborne InSAR. (**a**) InSAR platform in the ECR coordinate system, (**b**) platform coordinate system, and (**c**) baseline vector in the MMC system.

Figure 3a shows the geometric relation of airborne InSAR in the Earth Centered Rotating (ECR) coordinate system. $A_1$ and $A_2$ represent the position of the phase center of the main and slave antenna of the carrier, respectively, and H denotes the height of the phase center of the main antenna. $b$ represents the baseline vector from the main antenna to the slave antenna. P refers to the position of a ground target, and its elevation is h. The vector from the phase center of the main antenna to the ground target is $r$, and the unit vector in this direction is denoted by $\hat{r}$.

Figure 3b shows the SAR platform coordinate system and the orientation angles. The $X_P$-axis points in the ideal track direction, the $Z_P$-axis points in the zenith direction, and the $Y_P$-axis is perpendicular to the plane composed of the $X_P$-axis and $Z_P$-axis and points in the left direction. The carrier's orientation in flight is expressed by the roll angle $\theta_r$, pitch angle $\theta_p$, and yaw angle $\theta_y$. The rotation matrix is expressed as [18]:

$$R_r = \begin{bmatrix} 1 & 0 & 0 \\ 0 & \cos\theta_r & -\sin\theta_r \\ 0 & \sin\theta_r & \cos\theta_r \end{bmatrix} R_p = \begin{bmatrix} \cos\theta_p & 0 & -\sin\theta_p \\ 0 & 1 & 0 \\ \sin\theta_p & 0 & \cos\theta_p \end{bmatrix} R_y = \begin{bmatrix} \cos\theta_y & -\sin\theta_y & 0 \\ \sin\theta_y & \cos\theta_y & 0 \\ 0 & 0 & 1 \end{bmatrix}, \quad (1)$$

Figure 3c shows a decomposition of the baseline vector in the MMC system, where $b_v$ and $b_n$ refer to the cross-track and along-track components of the baseline, respectively. $\theta_b$ denotes the angle between $b$ and horizontal plane, while $\alpha$ denotes the angle between $b_n$ and horizontal plane. $V$ refers to the direction of the platform velocity vector, $W$ refers to the cross product of the velocity and baseline, and $N$ is determined by $W$ and $V$ following the right-hand rules. $\hat{v}, \hat{n}, \hat{w}$ are unit vectors on the three axes.

In the MMC system, the unit look vector can be represented by $\hat{v}, \hat{n}, \hat{w}$, shown as [18]

$$\hat{r} = \mu\hat{v} + \eta\hat{n} + \xi\hat{w}, \quad (2)$$

where

$$\mu = \sin\beta, \eta = \sin\theta_1 - \frac{b_v}{b_n}\sin\beta$$

$$\xi = -k_2\sqrt{\cos^2\beta - \left(\sin\theta_1 - \frac{b_v}{b_n}\sin\beta\right)^2} \quad (3)$$

and $\beta$ refers to the initial squint angle, and $\theta_1$ refers to the echo arrival azimuth, described as

$$\theta_1 = \arcsin(\frac{b^2}{2rb_n} + k_1k_2\frac{\lambda\varphi}{2Q\pi b_n} - \frac{\lambda^2\varphi^2}{8Q^2\pi^2 rb_n}).$$ (4)

where $\lambda$ refers to the wavelength of the signal and $\varphi$ refers to the phase difference between echoes on the two antennas. $Q$ is the InSAR operation mode index with $Q = 1$ for the standard mode and $Q = 2$ for the ping-pong mode. $k_1$ refers to the side-looking direction with $k_1 = 1$ as the right side observation and $k_1 = -1$ as the left side observation. $k_2$ refers to the location of the master antenna with $k_2 = 1$ for the right master antenna and $k_2 = -1$ for the left master antenna.

Let $\mathbf{\Gamma}$ denote the rotation matrix from the MMC system to the ground coordinate system, namely,

$$\mathbf{\Gamma} = \begin{bmatrix} 1 & 0 & 0 \\ 0 & k_2\cos\theta_{b_n} & k_1k_2\sin\theta_{b_n} \\ 0 & -k_1k_2\sin\theta_{b_n} & k_2\cos\theta_{b_n} \end{bmatrix} = \begin{bmatrix} 1 & 0 & 0 \\ 0 & k_2\sqrt{1-\frac{b^2}{b_n^2}\sin^2\theta_b} & \frac{k_1k_2b}{b_n}\sin\theta_b \\ 0 & -\frac{k_1k_2b}{b_n}\sin\theta_b & k_2\sqrt{1-\frac{b^2}{b_n^2}\sin^2\theta_b} \end{bmatrix}.$$ (5)

According to Equations (2) and (5), position $P$ can be expressed as

$$P = A + r = A + rR_yR_pR_r\mathbf{\Gamma}\hat{r}.$$ (6)

Under the condition of the broadside model, the center plane of the radar beam is a zero Doppler plane when the initial squint angle equals zero. Suppose the main antenna is on the left side of the carrier with the broadside model on the right, then the terrain elevation is expressed simply as

$$h = H - r(\cos\theta_p\cos(\alpha - \theta_r)\cos\theta_1 + \cos\theta_p\sin(\alpha - \theta_r)\sin\theta_1).$$ (7)

**4. Processing Procedure**

The main process of airborne InSAR data is divided into three parts, including interference processing within SLC scenes, global adjustment of interferometric parameters based on sparse GCPs and TPs between SLC scenes, and generation of a DSM and DOM based on the calibrated parameters. The processing flow chart for generating a DSM/DOM with airborne InSAR is shown in Figure 4.

*4.1. Interferometric Processing*

Interferometric processing is the first step in generating a DOM/DSM, mainly including registration, prefiltering, generation of interferograms, flat-earth effect removal, interferogram filtering, and PU. In this section, prefiltering and flat-earth effect removal are ignored in the processing process.

4.1.1. Registration and Generation of Interferograms

In generating a DOM/DSM, registration is the most basic and critical step. Its core idea is determining the relative offset of the matching position between two SAR complex images. The imprecision of SAR image registration leads to large interferometric phase errors. There is a popular supposition that a registration accuracy of 1/10 pixels has no significant influence on the quality of the interferogram [20]. There are three common matching measures based on complex similarity, phase similarity, and intensity similarity [21]. We take the complex correlation coregistration based on FFT to obtain the exact offset of homologous points. After polynomial fitting of the matching data, the coordinate conversion relation of the master and slave SLC pairs is calculated, which can be used to complete the resampling of the slave image [22]. Finally, the complex values of the master image and slave SLC pairs are in conjugate multiplication to calculate the interferogram.

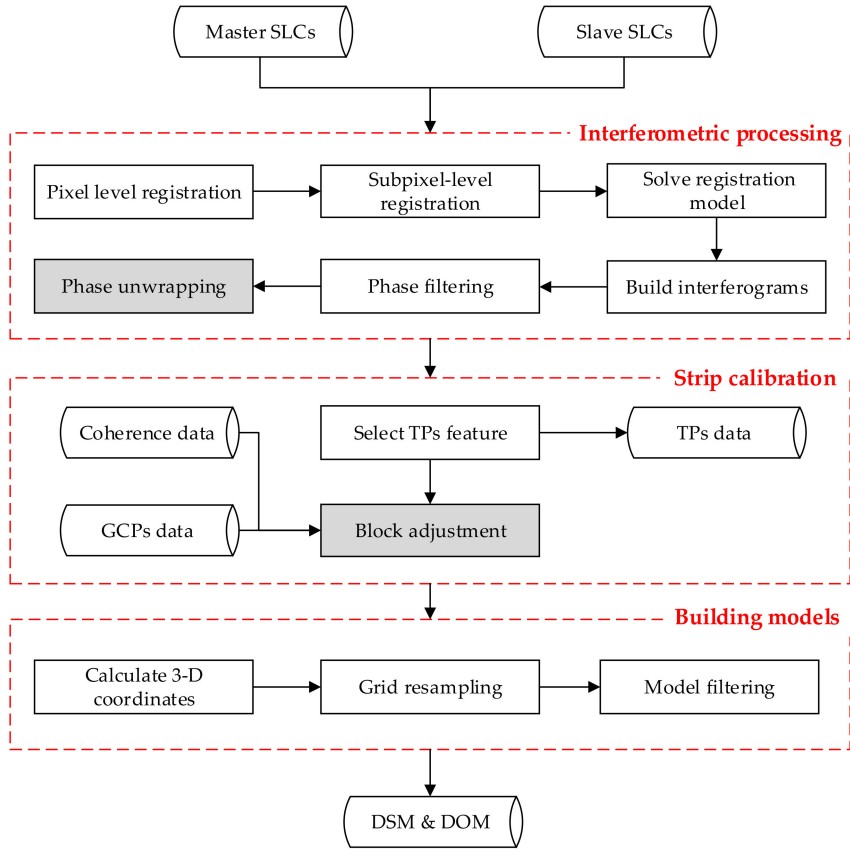

**Figure 4.** Processing flow chart of a DOM and DSM from airborne InSAR.

### 4.1.2. Interferogram Filtering

Due to the inherent speckle noise effect of SAR (thermal noise of the radar system, radar shadow, registration error, etc.), there are many phase noises in the interferogram, which are manifested as discontinuous phases. The phase periodicity and fringe pattern are not obvious, making a large impact on subsequent unwrapping [22–24]. The commonly used filtering methods include spatial domain filtering, frequency domain filtering, and time-frequency analysis filtering. In our experiment, we utilized Goldstein filtering [25] to filter the interferogram. After the overlapping phase, blocks were selected and processed in the frequency domain using a smoothing filter, and we used the filtering parameters to process the power spectrum.

### 4.1.3. Unwrapping Method

The phase values of the interferogram obtained by interferometric processing were wrapped, and their range was [0, 2π] or [−π, π]. To obtain accurate terrain elevation information, the absolute interferometric phase values corresponding to the interferogram needed to be restored. PU uses a certain mathematical method to calculate the number of integer cycles of the phase difference between each interferometric phase to obtain the continuous distribution of the interferometric phase [25]. We utilized a fast and accurate unwrapping method based on minimum balanced trees (MBTs), which can be executed in parallel mode and has better efficiency. The MBT method builds minimum balanced trees and calculates unwrapping priority sequence maps to execute phase wrapping [26,27].

Implementation of the unwrapping method focuses on constructing reliability maps and MBTs on discrete samples. There are two types of points, one for data sampling positions and another for potential phase discontinuity boundaries. The edges that connect the boundary points are potential discontinuity boundaries. Each boundary point is designed with attributes of reliability, source seeds, and priority. Each boundary edge has a quality weight and a difference across forwarding and reverse-pair connecting lines.

Sampling connecting edges are potential integration paths. Each sampling point has a wrapped phase, integration priority, and unwrapped phase, and each sampling edge has a wrapped phase gradient used in the path integration. The unwrapping processing flow is described briefly in Figure 5.

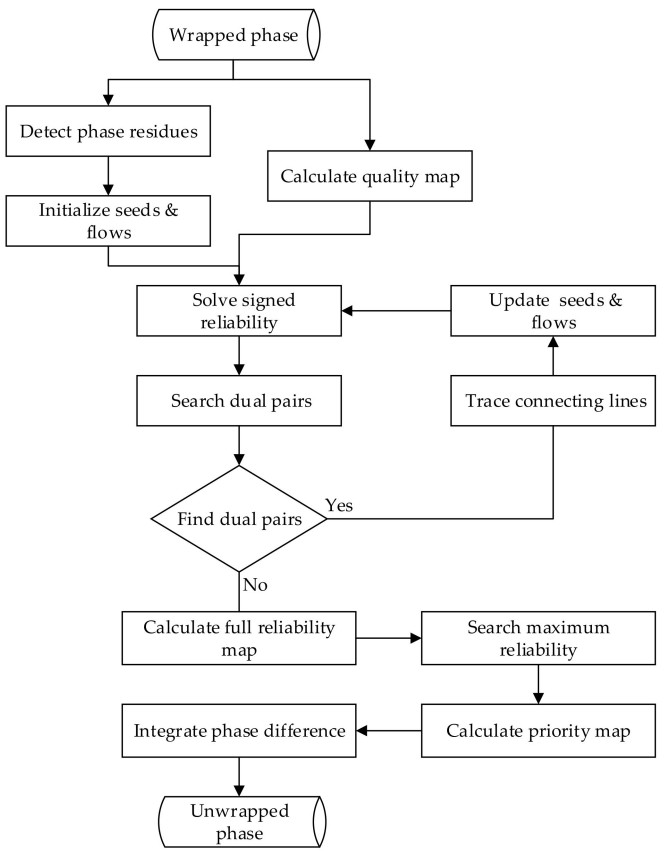

**Figure 5.** Processing flow chart of PU with MBTs.

The reliability, $p_+$ and $p_-$, at residue points is zero, and the relation can be defined in Eikonal equations as

$$|\Delta p_+(m,n)| = |\Delta p_-(m,n)| = q(m,n). \tag{8}$$

where $|\cdot|$ is the absolute value operator, $\Delta$ is the difference operator, and $q$ refers to the quality weight. There are different ways to define the quality weight $q$, including constants, pseudocoherence coefficient, the inverse of maximum gradient, or derivative variance [28]. Namely,

$$q = \begin{cases} 1 & \text{constants,} \\ c_p & \text{pseudo coherence,} \\ 1/\sigma_\Delta^2 & \text{derivative variance,} \\ 1/\max(|\Delta|) & \text{maximum gradien.} \end{cases} \tag{9}$$

where $c_p$ refers to the pseudo coherence coefficient, $\sigma_\Delta^2$ refers to the derivative variance including two directions, and $\max(|\Delta|)$ refers to the maximum absolute value of the gradient. Moreover, most processing in Figure 5 is designed to run in parallel. When big data blocks are divided into small blocks as individual computing units, the proposed method achieves higher efficiency.

### 4.2. Strip Calibration

#### 4.2.1. Error Sensitivity Analysis

As shown in Equation (7), the system parameters that influence the positioning accuracy of the DEM include the height of the master antenna phase center ($H$), slant range ($r$), baseline length ($b$), baseline angle ($\alpha$), absolute interferometric phase ($\varphi$), roll angle ($\theta_r$), and pitch angle ($\theta_p$). We represent the errors of these parameters with d$h$, d$H$, d$r$, d$b$, d$\alpha$, d$\varphi$, d$\theta_r$, and d$\theta_p$ and assume that these parameters are independent of each other, according to the covariance propagation law. The height error variance of the DEM can be described with the differential coefficient as [18]

$$\sigma_h^2 = (\frac{\partial h}{\partial H})^2\sigma_H^2 + (\frac{\partial h}{\partial r})^2\sigma_r^2 + (\frac{\partial h}{\partial b})^2\sigma_b^2 + (\frac{\partial h}{\partial \alpha})^2\sigma_\alpha^2 + (\frac{\partial h}{\partial \varphi})^2\sigma_\varphi^2 + (\frac{\partial h}{\partial \theta_r})^2\sigma_{\theta_r}^2 + (\frac{\partial h}{\partial \theta_p})^2\sigma_p^2, \quad (10)$$

where

$$\frac{\partial h}{\partial H} = 1, \quad (11)$$

$$\frac{\partial h}{\partial r} = -\cos\theta + l(\frac{\lambda^2\varphi^2}{8\pi^2rb^2} - \frac{b}{2r}), \quad (12)$$

$$\frac{\partial h}{\partial b} = rl(\frac{1}{2r} + \frac{\lambda\varphi}{2\pi b^2} + \frac{\lambda^2\varphi^2}{8\pi^2rb^2}), \quad (13)$$

$$\frac{\partial h}{\partial \alpha} = rl\sqrt{\cos^2\beta - \sin^2\theta_1}, \quad (14)$$

$$\frac{\partial h}{\partial \varphi} = rl(-\frac{\lambda}{2\pi b} - \frac{\lambda^2\varphi}{4\pi^2rb}), \quad (15)$$

$$\frac{\partial h}{\partial \theta_r} = -rl\sqrt{\cos^2\beta - \sin^2\theta_1}, \quad (16)$$

$$\frac{\partial h}{\partial \theta_p} = r[\sin\theta_p\cos(\alpha - \theta_r)\sqrt{\cos^2\beta - \sin^2\theta_1} - \sin\theta_p\sin(\alpha - \theta_r)\sin\theta_1], \quad (17)$$

and $l$ can be expressed as

$$l = \cos\theta_p(\frac{\cos(\alpha - \theta_r)\sin\theta_1 + \sin(\alpha - \theta_r)\cos\theta_1}{\cos\theta_1}), \quad (18)$$

Under specific conditions, the influence level caused by the errors might be different. The coefficient of the above expressions provides a basis for analyzing the error sensitivity.

The roll angle error and baseline angle error are combined into one during processing. The constant difference between the unwrapped phase and real phase, namely, the phase offset, must be correctly calibrated. Therefore, it is necessary to calibrate the baseline, baseline angle, and phase offset of all interferometric data with the POS data of the InSAR carrier, GCPs with known 3D coordinates, and TPs in the overlap area.

#### 4.2.2. Block Adjustment

Based on the SIFT algorithm [29,30], the TPs in the adjacent strip overlapping in the experimental areas are automatically selected. The coherence coefficient of selected TPs is expected to be greater than 0.9, and the bias is less than 0.5 pixels in the strips. In addition, the thresholds are 0.8 and 3 pixels between the strips. Afterward, we can carry out block adjustment [31–33] to calibrate the system parameters and calculate the coordinates of the TPs.

With known coordinates of GCPs, Equation (6) can be rewritten as

$$\begin{aligned} X_G &= F_X^i(b^i, \alpha^i, \varphi^i) \\ Y_G &= F_Y^i(b^i, \alpha^i, \varphi^i) \\ Z_G &= F_Z^i(b^i, \alpha^i, \varphi^i) \end{aligned} \quad (19)$$

where $i$ refers to the index of interferogram data. For a certain TP, its three-dimensional coordinates should be consistent within its two adjacent interferogram areas. The coordinate differences, denoted with $X_{\Delta T}$, $Y_{\Delta T}$, and $Z_{\Delta T}$, are taken as new three-dimensional coordinates with zero values. The observation equation is described as

$$
\begin{aligned}
X_{\Delta T}^{ij} &= \mathrm{F}_X^i(b^i, \alpha^i, \varphi^i) - \mathrm{F}_X^j(b^j, \alpha^j, \varphi^j) \\
Y_{\Delta T}^{ij} &= \mathrm{F}_Y^i(b^i, \alpha^i, \varphi^i) - \mathrm{F}_Y^j(b^j, \alpha^j, \varphi^j) \\
Z_{\Delta T}^{ij} &= \mathrm{F}_Z^i(b^i, \alpha^i, \varphi^i) - \mathrm{F}_Y^j(b^j, \alpha^j, \varphi^j)
\end{aligned}
\tag{20}
$$

Equations (19) and (20) are nonlinear equations, which are linearized according to the Taylor formula for a certain GCP or TP, shown as

$$
\begin{bmatrix} V_{GX}^i \\ V_{GY}^i \\ V_{GZ}^i \end{bmatrix} =
\begin{bmatrix}
\frac{\partial F_X^i}{\partial b^i} & \frac{\partial F_X^i}{\partial \alpha^i} & \frac{\partial F_X^i}{\partial \varphi^i} \\
\frac{\partial F_Y^i}{\partial b^i} & \frac{\partial F_Y^i}{\partial \alpha^i} & \frac{\partial F_Y^i}{\partial \varphi^i} \\
\frac{\partial F_Z^i}{\partial b^i} & \frac{\partial F_Z^i}{\partial \alpha^i} & \frac{\partial F_Z^i}{\partial \varphi^i}
\end{bmatrix}
\begin{bmatrix} \delta b^i \\ \delta \alpha^i \\ \delta \varphi^i \end{bmatrix} -
\begin{bmatrix} L_{GX}^i \\ L_{GY}^i \\ L_{GZ}^i \end{bmatrix},
\tag{21}
$$

$$
\begin{bmatrix} V_{\Delta TX}^{ij} \\ V_{\Delta TY}^{ij} \\ V_{\Delta TZ}^{ij} \end{bmatrix} =
\begin{bmatrix}
\frac{\partial F_X^i}{\partial b^i} & \frac{\partial F_X^i}{\partial \alpha^i} & \frac{\partial F_X^i}{\partial \varphi^i} & -\frac{\partial F_X^j}{\partial b^j} & -\frac{\partial F_X^j}{\partial \alpha^j} & -\frac{\partial F_X^j}{\partial \varphi^j} \\
\frac{\partial F_Y^i}{\partial b^i} & \frac{\partial F_Y^i}{\partial \alpha^i} & \frac{\partial F_Y^i}{\partial \varphi^i} & -\frac{\partial F_Y^j}{\partial b^j} & -\frac{\partial F_Y^j}{\partial \alpha^j} & -\frac{\partial F_Y^j}{\partial \varphi^j} \\
\frac{\partial F_Z^i}{\partial b^i} & \frac{\partial F_Z^i}{\partial \alpha^i} & \frac{\partial F_Z^i}{\partial \varphi^i} & -\frac{\partial F_Z^j}{\partial b^j} & -\frac{\partial F_Z^j}{\partial \alpha^j} & -\frac{\partial F_Z^j}{\partial \varphi^j}
\end{bmatrix}
\begin{bmatrix} \delta b^i \\ \delta \alpha^i \\ \delta \varphi^i \\ \delta b^j \\ \delta \alpha^j \\ \delta \varphi^j \end{bmatrix} -
\begin{bmatrix} L_{\Delta TX}^{ij} \\ L_{\Delta TY}^{ij} \\ L_{\Delta TZ}^{ij} \end{bmatrix}.
\tag{22}
$$

where $V$ is the residual of coordinates; $\delta b$, $\delta \alpha$ and $\delta \varphi$ are the corrections of interferometric parameters; and $L$ is a constant item.

Based on the principle of least squares, the corrections of the three-dimensional coordinates of the TPs are carried out first when solving the normal equation to obtain corrections of the interferometric parameters of the system. Then, we apply the parameter corrections, and the coordinates of the TPs are solved. The model parameter is solved with an iterative and gradual scheme. The process of block adjustment is shown in Figure 6:

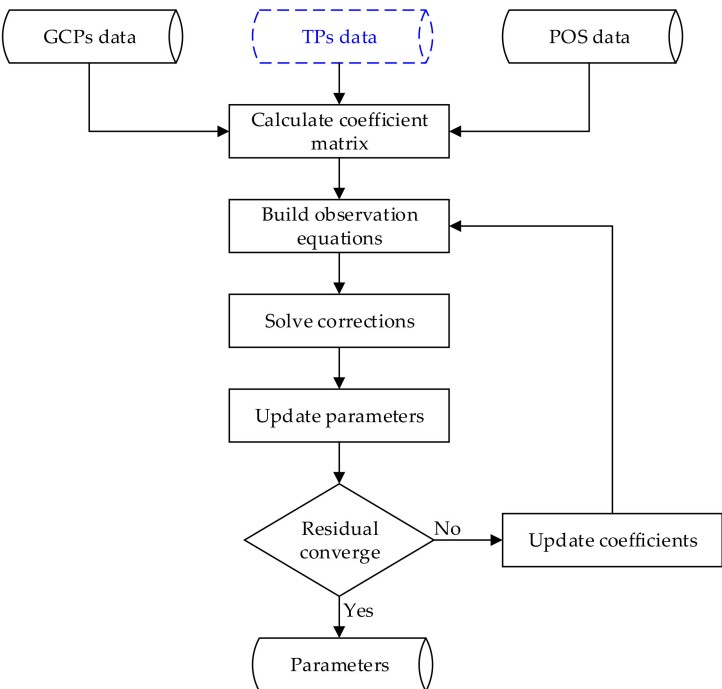

**Figure 6.** The processing flow of block adjustment.

A small number of GCPs and a large number of TPs are utilized to calibrate the system parameters of Heyang, Shibing, and Qionglai in block adjustment, and we applied the system parameters corresponding to all interferometric data in all three areas.

### 4.3. Building Models

As shown in Equation (6), we calculated the coordinates $X$, $Y$, and $H$ of every data sample of each area, which formed three-dimensional point clouds. With the set resolution of the resulting DOM/DSM (0.5 m) and a coherence threshold (0.6), the DOM and DSM were built by grid interpolation. We obtained the DOM with a resolution of 0.5 m and the DSM with a grid spacing of 0.5 m. All DOMs and DSMs in the experimental areas were spliced and inlaid. If there were significant differences near the spliced joint after splicing, the DOM values were filtered and smoothed.

With the procedure, we implemented a self-developed airborne SAR data-processing tool, AirborneInSARMap. Applying a parallel-processing scheme, the corresponding processing of each SLC and interferogram could be run in batches for efficiency and reliability.

## 5. Results
### 5.1. Results of Interferometric Processing

Taking into account the following regional adjustment, the adjacent interferometric scenes maintained a certain overlap. Goldstein filtering improved the continuity quality of the interferometric phase. The 13 scenes of interferometric phase data obtained in the Qionglai experiment, which cover undulating mountains, are shown in Figure 7.

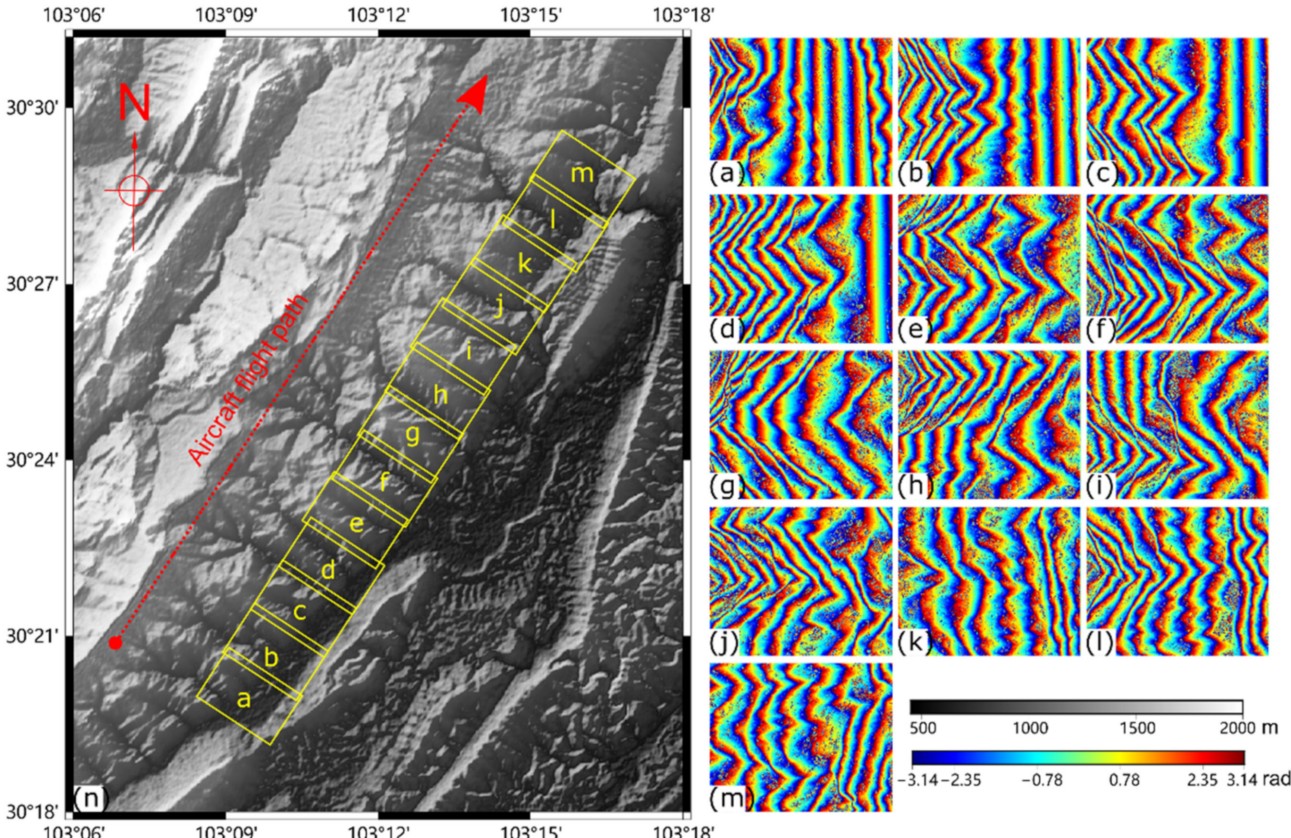

**Figure 7.** Scene coverage distribution of the Qionglai experiment (**n**), where a 30-m resolution DEM is taken as the base map, and filtered interferometric phase of the 13 scenes (**a–m**).

However, phase filtering cannot reduce the challenge of interfering PU in rugged mountainous areas. The overlap and shadows in mountainous areas caused the interference fringes to be severely distorted, and the two-dimensional continuity was destroyed there.

The 13 obtained scenes of unwrapped results from the proposed PU method based on MBTs are shown in Figure 8. The proposed unwrapping method brought about stable and reliable unwrapped results in areas of heavy terrain undulations.

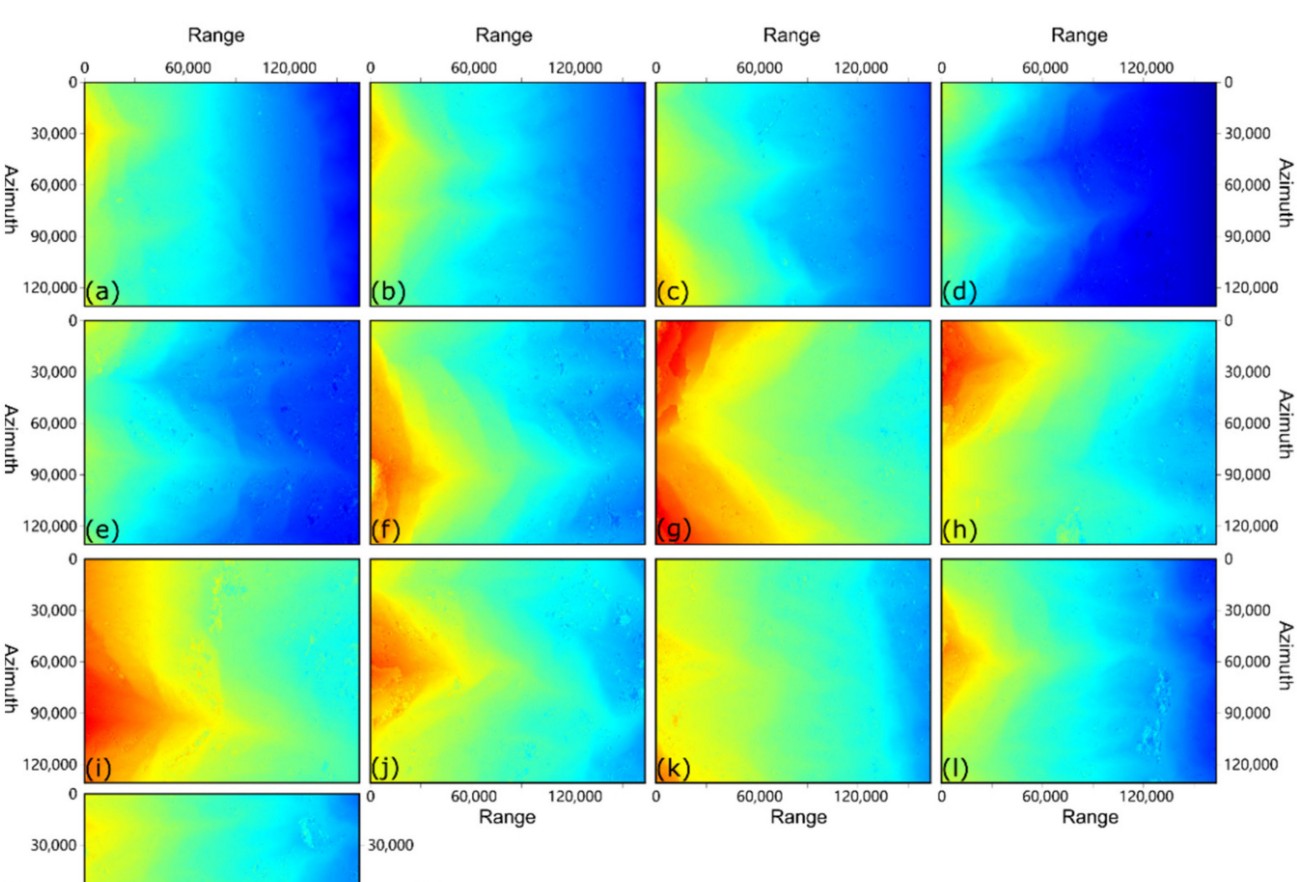

**Figure 8.** The 13 scenes of unwrapped phase data in the Qionglai experiment from the proposed PU method based on MBTs, (**a**–**m**) refer to unwrapped phase in scene **a**–**m** in Figure 7.

### 5.2. Results of Height Error Sensitivity Analysis

Sensitivity expressions in Equations (12)–(17) involve different variables, especially those that change dynamically. When calculating the sensitivity, certain reference values are needed. For example, the orientation angles, whose average values are taken for the SAR imaging range. For example, when reference $\theta_r$, $\theta_p$, and $\theta_y$ are set as 0.023°, 0.649°, and 2.254°, respectively, the look angle range is 20°~70°, and the master antenna phase center height and ground reference height are 4043.116 m and 900 m, respectively. The height error sensitivity analysis of slant range ($r$), baseline length ($b$), baseline angle ($\alpha$), absolute interferometric phase ($\varphi$), roll angle ($\theta_r$), and pitch angle ($\theta_p$) are calculated, as shown in Figure 9a–f.

As shown in Figure 9, the sensitivity coefficients of the baseline length, baseline angle, and roll angle are approximately $n \times 10^3$ with increasing look angles, far greater than those of the other parameters. Since the aircraft carries POS, the height of the master antenna phase center and the slant range can be accurately calculated, and the pitch angle error is of little effect and is ignorable in a single scene.

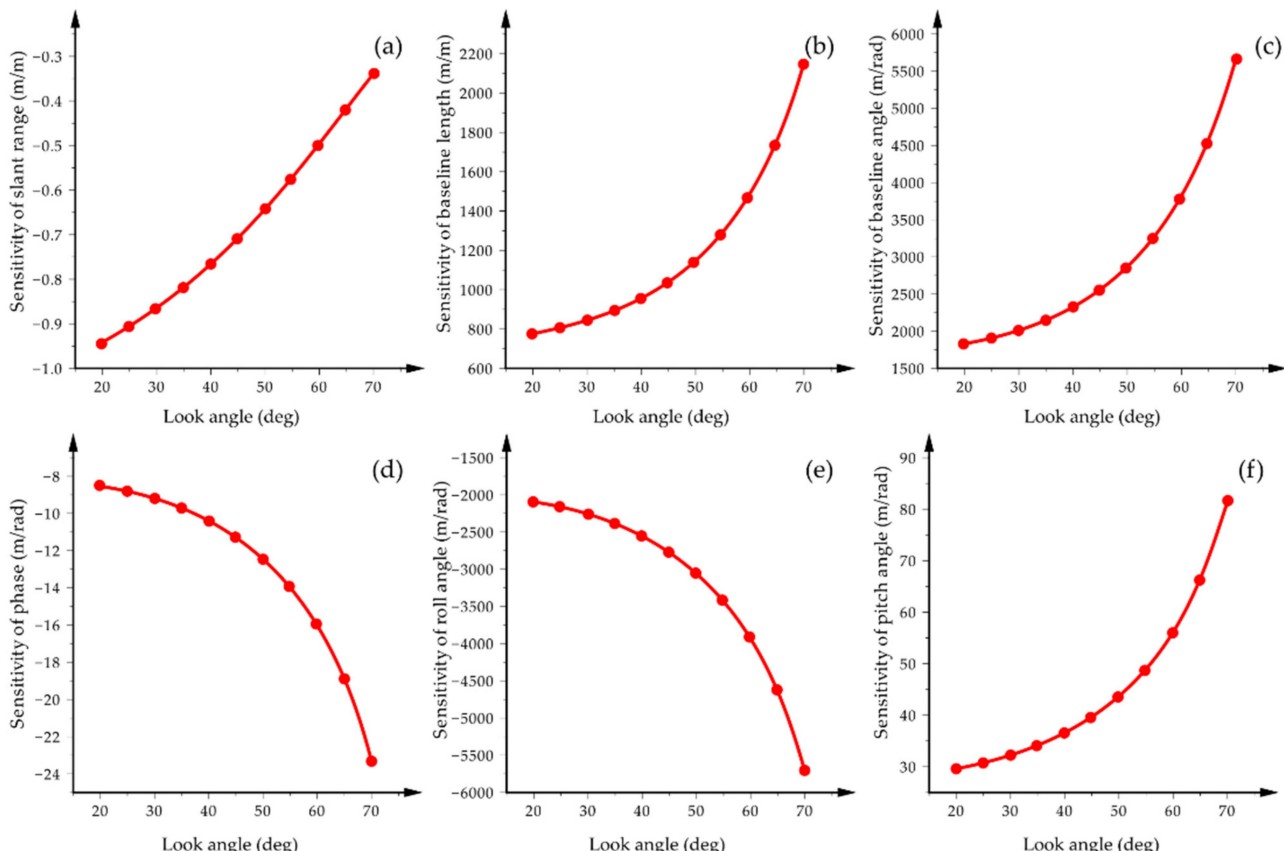

**Figure 9.** Height error sensitivity of (**a**) slant range, (**b**) baseline length, (**c**) baseline angle, (**d**) interferometric phase, (**e**) roll angle, and (**f**) pitch angle with given parameters for a single scene.

### 5.3. Results of Strip Calibration

We calibrated the interferometric parameters of interferograms with GCPs (9 in the Heyang area, 14 in the Shibing area, and 12 in the Qionglai area) according to Equation (21). After several iterations, all parameters reach convergence. The calibration results of the first interferograms in the Qionglai area are shown in Table 2. The data in the table show there are biases between the antenna baseline length and nominal value, and the difference is no more than 1 cm. The baseline angle values have a larger variation range because of a combination of the dynamic roll angle error.

**Table 2.** Parameter calibration of the first scene of the Qionglai experiment in iterations.

| Iteration | $b$ (m) | $\alpha$ (rad) | $\varphi_o$ (rad) |
|---|---|---|---|
| 0 | 0.313000 | 0.912372 | 32.5086 |
| 1 | 0.314680 | 0.848153 | 17.8841 |
| 2 | 0.315350 | 0.847576 | 17.6513 |
| 3 | 0.315352 | 0.847562 | 17.6477 |
| 4 | 0.315352 | 0.847563 | 17.6478 |
| 5 | 0.315352 | 0.847563 | 17.6478 |
| 6 | 0.315352 | 0.847563 | 17.6479 |
| 7 | 0.315352 | 0.847562 | 17.6477 |
| 8 | 0.315352 | 0.847563 | 17.6479 |
| 9 | 0.315352 | 0.847563 | 17.6478 |
| 10 | 0.315352 | 0.847563 | 17.6478 |

More than 10 TPs in each adjacent scene overlap in the experimental areas selected automatically with the SIFT algorithm. The TPs and their connections in the first 4 scenes are shown in Figure 10.

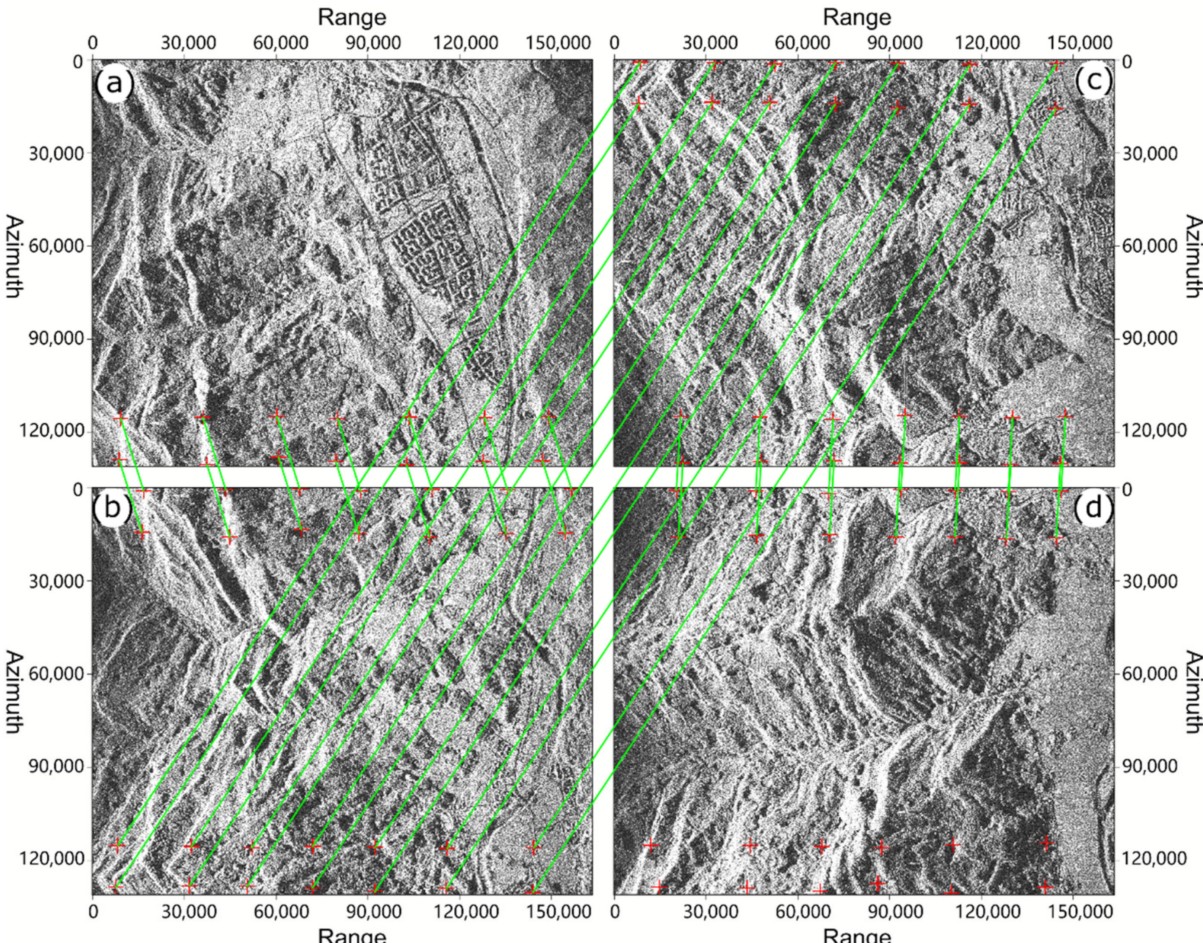

**Figure 10.** TPs (red crosses) and their connections (green lines) in the first 4 scenes of the Qionglai experiment, where (**a**–**d**) refer to the magnitude maps of the 4 scenes.

The GCPs and TPs were all taken during block adjustment of the parameters for strip calibration. The parameter solutions in the processing all converge in several iterations, and the obtained values in the Qionglai experiment are shown in Table 3.

**Table 3.** Parameter adjustment results of the Qionglai experiment.

| Scene | $b$ (m) | $\alpha$ (rad) | $\varphi_0$ (rad) |
|---|---|---|---|
| b | 0.31449 | 0.912298 | 32.5846 |
| c | 0.31430 | 0.912200 | 32.6022 |
| d | 0.31388 | 0.912538 | 51.4157 |
| e | 0.31394 | 0.911728 | 39.1988 |
| f | 0.31392 | 0.911308 | 20.3440 |
| g | 0.31404 | 0.911853 | 0.9472 |
| h | 0.31507 | 0.911262 | 13.7681 |
| i | 0.31453 | 0.911440 | 1.1327 |
| j | 0.31260 | 0.911610 | 13.7511 |
| k | 0.31322 | 0.911293 | 13.9237 |
| l | 0.31269 | 0.911433 | 26.3863 |
| m | 0.31407 | 0.911785 | 20.1145 |

*5.4. Results of the DOM and DSM*

After calculating the three-dimensional points of the interferogram, DOM and DSM data of the three experimental areas were obtained by grid interpolation with a resolution of 0.5 m. The geocoded magnitude map and height map of the first scene in the Qionglai

experiment are shown in Figure 11. For better analysis and comparison, smoothed height contours and a 30 m-resolution SRTM DEM are utilized in the DSM figures.

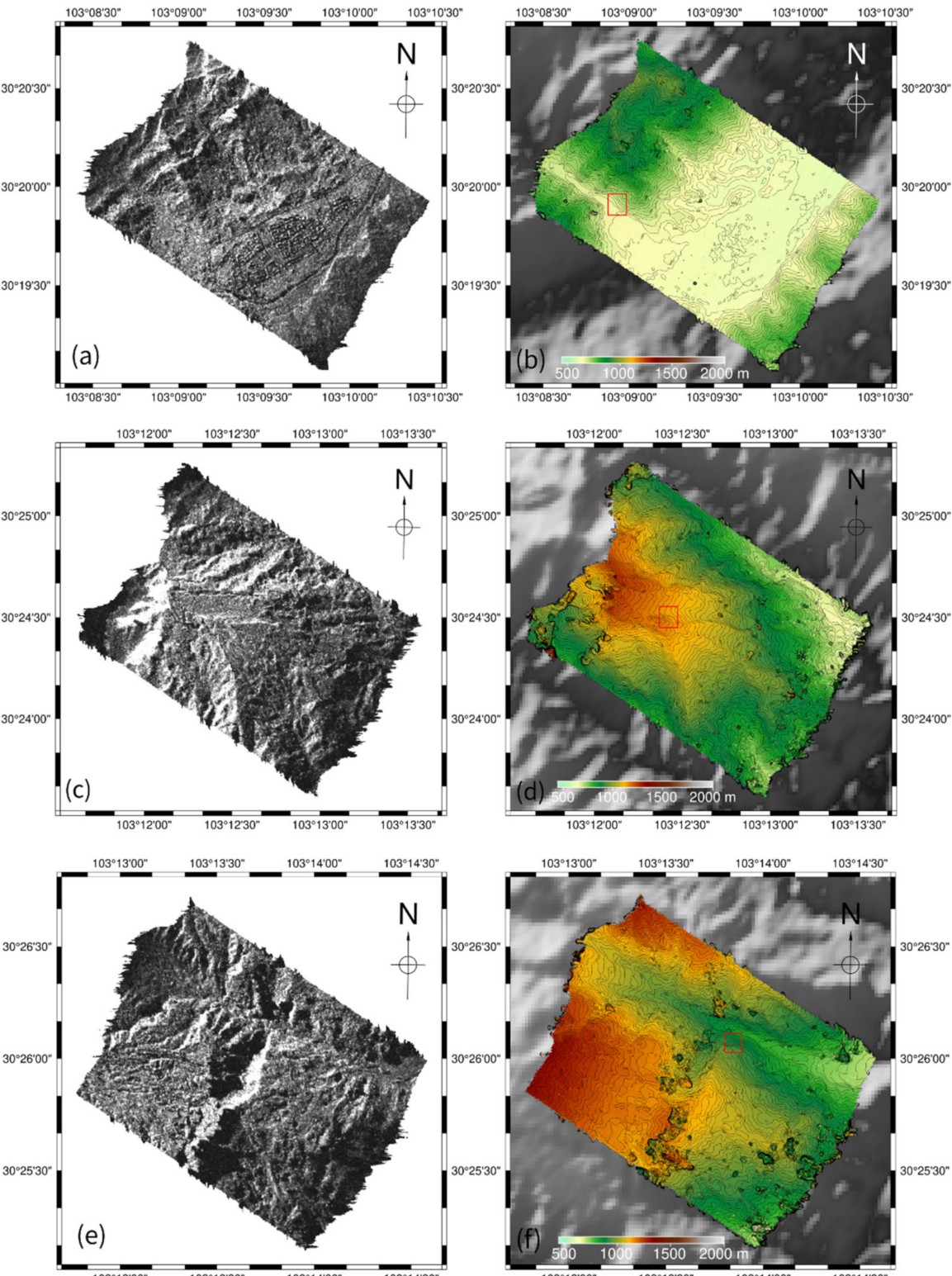

**Figure 11.** Geocoded scene magnitude maps and height maps of the Qionglai experiment. (**a,b**) refer to the magnitude map and height map of the first scene, (**c,d**) refer to the seventh scene, (**e,f**) refer to the ninth scene. The squares in (**b,d,f**) mark detail analysis areas.

The scene in Figure 11a,b covers the residential area of a town on the east side. To the northwest of the town is a mountainous area several hundred meters high. The distribution of the feature content is provided in the magnitude map, and the elevation change is provided in a more accurate description via the height map. The obtained seventh scene data of the Qionglai experiment are shown in Figure 11c,d, where upside slope shortening and back slope stretching appear in the magnitude map. The height map in Figure 11d provides an accurate description. A more challenging case appears in the ninth scene data, with large drops and steep ridges, as shown in Figure 11e,f. The merged DOM and DSM results of the three regions are shown in Figures 12–14, which demonstrate the reconstructed details of the airborne InSAR topographic modeling results.

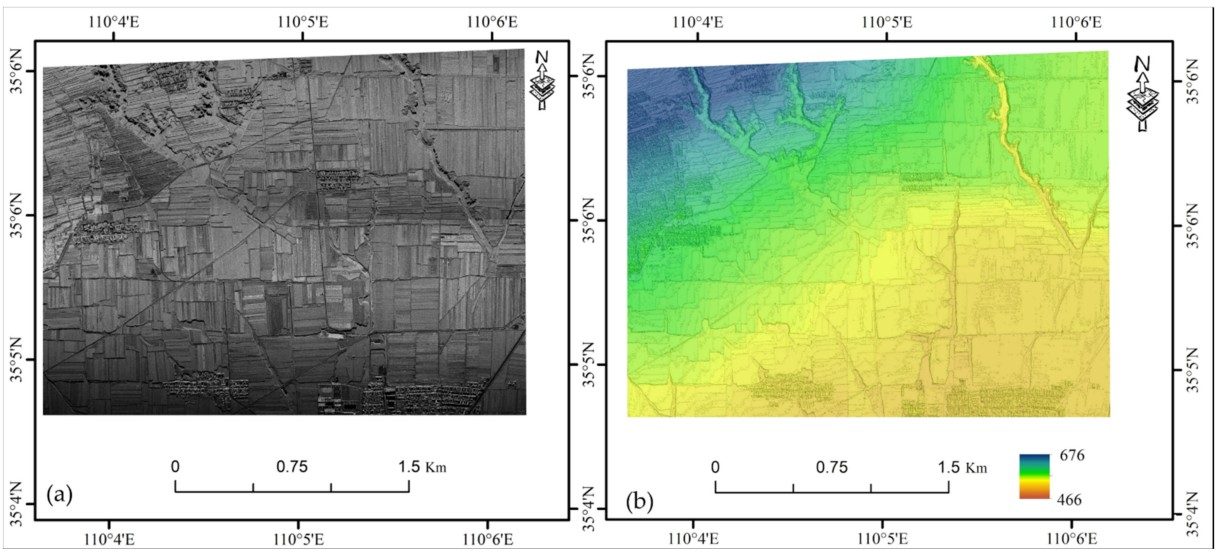

**Figure 12.** The obtained DOM (**a**) and DSM (**b**) of the Heyang experimental area.

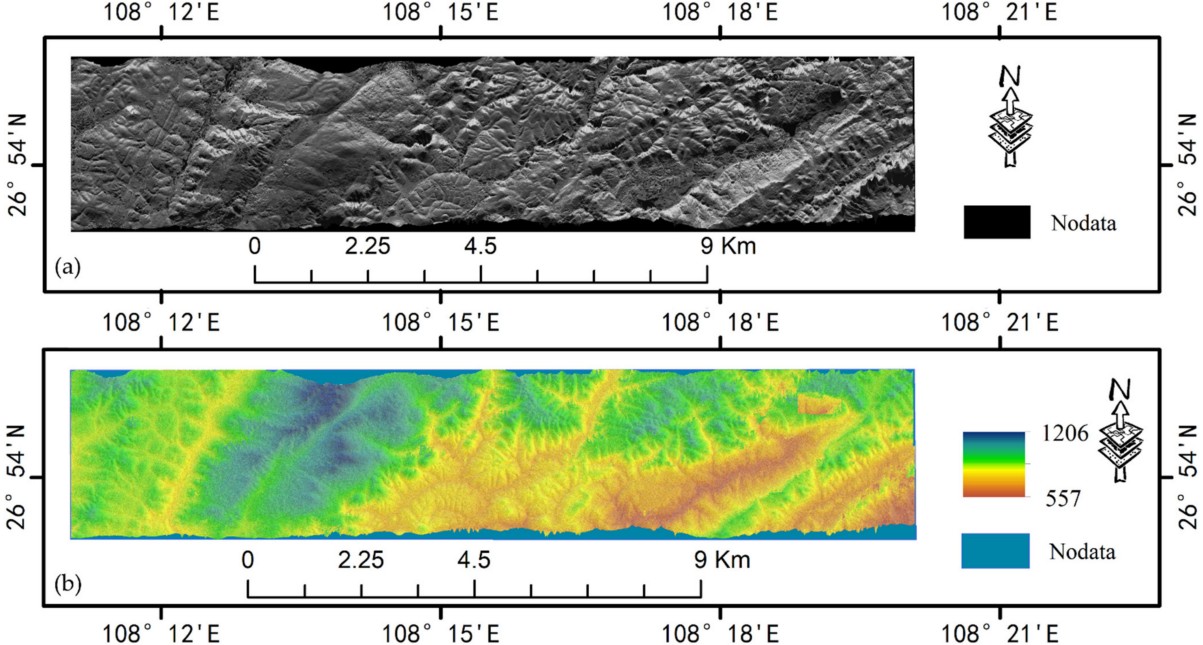

**Figure 13.** The obtained DOM (**a**) and DSM (**b**) of the Shibing experimental area.

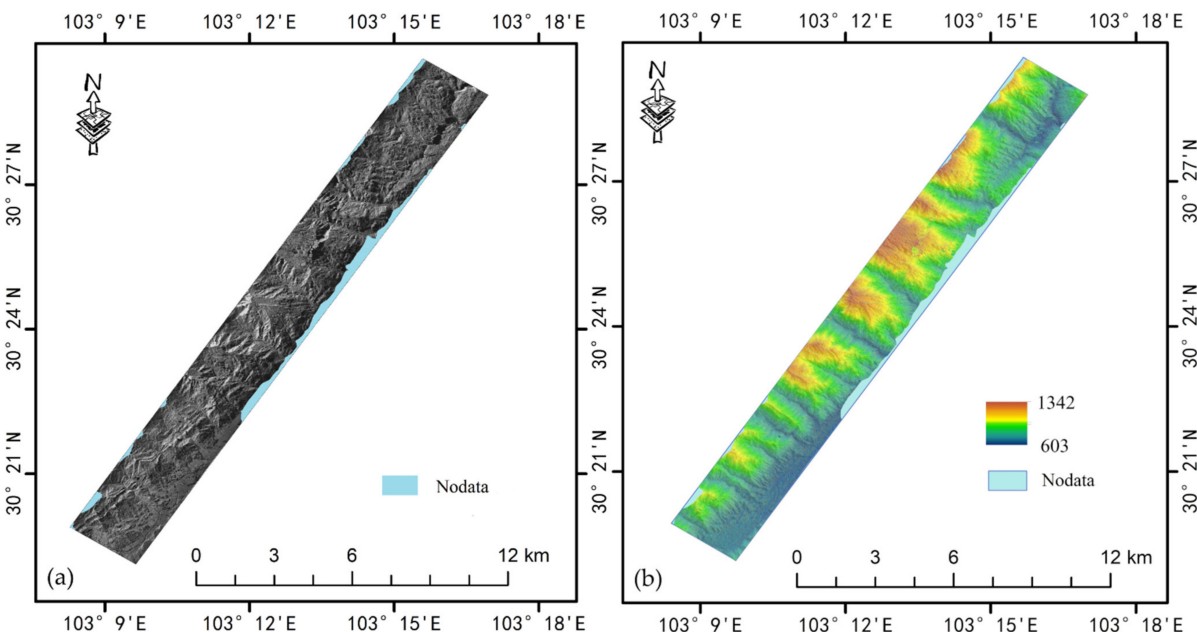

**Figure 14.** The obtained DOM (**a**) and DSM (**b**) of the Qionglai experimental area.

*5.5. Results of Accuracy Assessment*

For local detail assessment of the obtained DSM, three window areas were selected, as marked in Figure 11b,d,f. The first area was located at the foot of a mountain, covering a gentle slope and flat land; the second area was at the top of a mountain, covering peaks, valleys, and ridges; and the third area was on a steep slope of a mountain. We extracted the corresponding DSM data to plot 5 m-interval contour lines and compared them with those from the SRTM data. As shown in Figure 15a, our DSM data provide more local detailed descriptions, with obvious bumps on the slope and more clear horizontal features in the flat area in the first area. Figure 15b presents a finer description of the mountain than the smooth contours of the SRTM, and a more detailed description of the slope compared to SRTM in Figure 15c.

In consideration of the frequency spectrum analysis of DSM, we perform Fourier transform on the data of the second window area, 614 × 614 pixels. The SRTM DEM data are resampled due to differences in resolution. The frequency magnitude spectrum is shown in Figure 16. We found that the energy is more concentrated in the low-frequency portion for the SRTM data, while our DSM data have a more dispersive spectrum for their finer details.

For geometric accuracy assessment, we performed model accuracy verification with GCPs. 13, 12, and 13 GCPs were used to calculate local horizontal coordinate errors and elevation errors in the *x*, *y,* and *h* directions in the three experimental areas. The error distribution in the three experimental areas is shown in Figure 17.

To quantify the evaluation accuracy, the root mean square error (*RMSE*), mean absolute error (*MAE*), and standard deviation (*SD*) are taken. Those of height are defined as

$$RMSE_h = \sqrt{\frac{1}{n}\sum_{i=1}^{n}\left(h^i - h_G^i\right)^2}, \tag{23}$$

$$MAE_h = \frac{1}{n}\sum_{i=1}^{n}\left|h^i - h_G^i\right|, \tag{24}$$

$$SD_h = \sqrt{\frac{1}{n}\sum_{i=1}^{n}\left(h^i - h_G^i - \mu\right)^2}, \tag{25}$$

where $\mu$ denotes the average of $h - h_{\mathrm{G}}$ which can be defined as

$$\mu = \frac{1}{n}\sum_{i=1}^{n}\left(h^{i} - h_{G}^{i}\right). \tag{26}$$

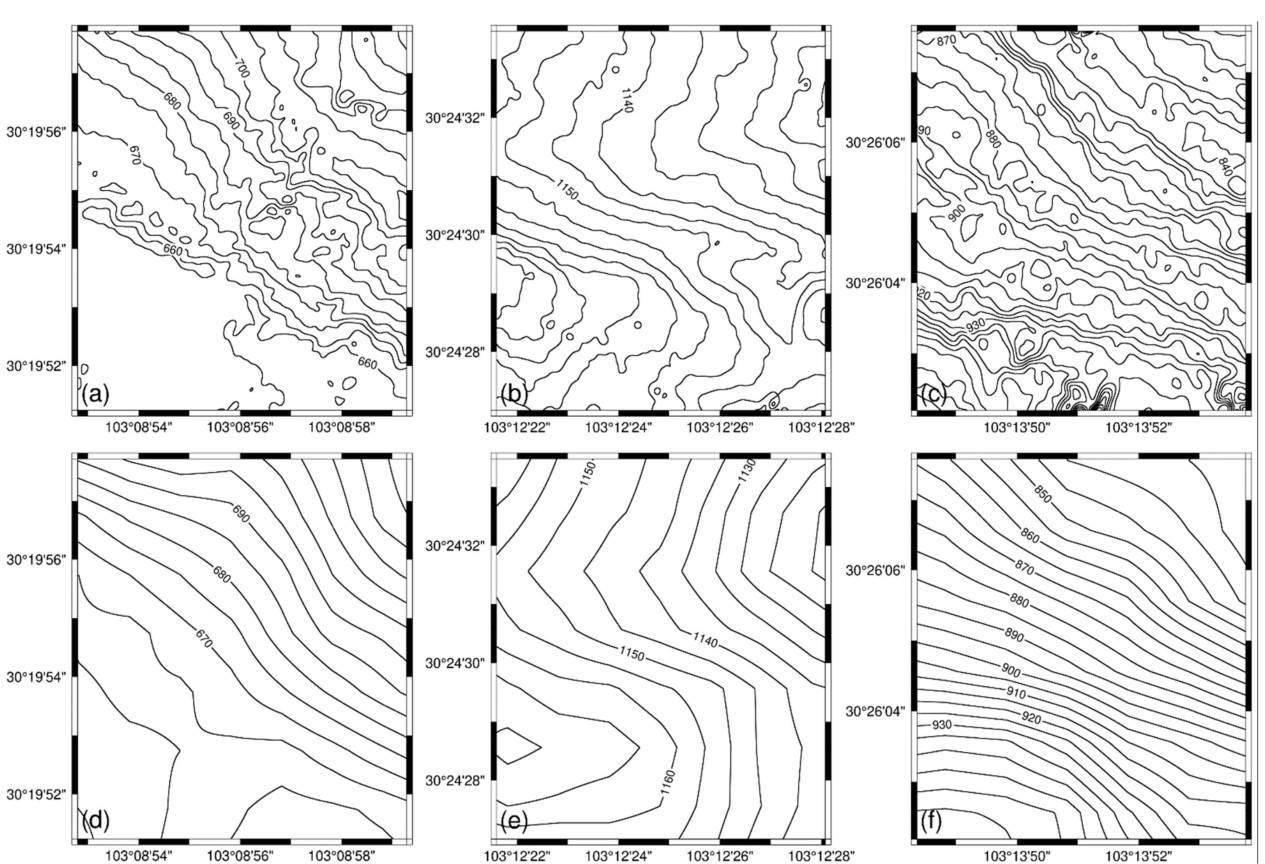

**Figure 15.** Contours of the windowed areas marked in Figure 11 from the obtained DSM (top) and SRTM DEM (bottom). (**a,d**) refer to Figure 11b, (**b,e**) refer to Figure 11d and (**c,f**) refer to Figure 11f.

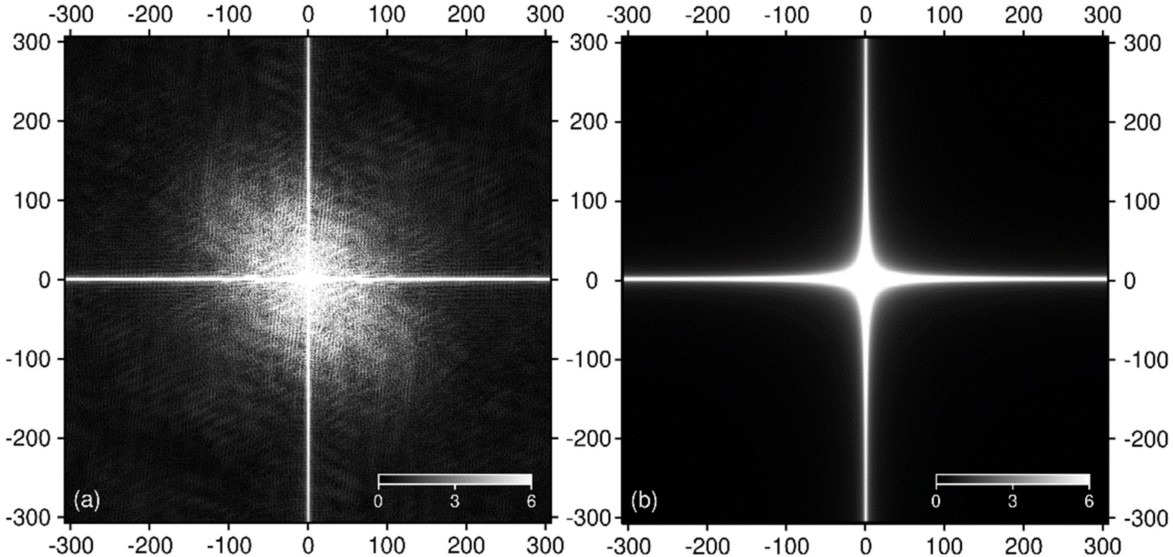

**Figure 16.** The frequency spectrum of the second windowed area marked in Figure 11 from the obtained DSM (**a**) and SRTM DEM (**b**).

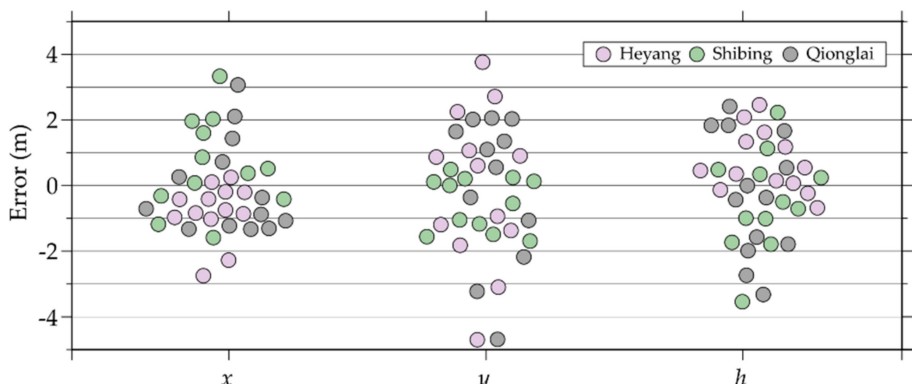

**Figure 17.** Error distribution of the *x*, *y*, and *h* directions in the Heyang experimental area in Shaanxi, Shibing experimental area in Guizhou, and Qionglai experimental area in Sichuan.

The definitions in the other two directions are similar to those of height. The statistical values of RMSE for the three regions are listed in Table 4.

**Table 4.** Accuracy assessment of DSMs results in *RMSE*, *MAE*, and *SD* (m) of the three experimental areas.

| Index | Direction | Heyang | Shibing | Qionglai |
|---|---|---|---|---|
| | *x* | 1.224 | 1.490 | 1.410 |
| *RSME* | *y* | 2.120 | 0.938 | 2.156 |
| | *h* | 1.150 | 1.433 | 1.846 |
| | *x* | 0.926 | 1.175 | 1.216 |
| *MAE* | *y* | 1.851 | 0.720 | 1.858 |
| | *h* | 0.876 | 1.073 | 1.580 |
| | *x* | 0.866 | 1.369 | 1.409 |
| *SD* | *y* | 2.543 | 0.771 | 2.154 |
| | *h* | 1.790 | 1.393 | 1.820 |

Table 4 shows the accuracy assessment results of topographic modeling in the three experimental areas. The RMSE is 1.490 m; 682 m, the RMSE is 1.464 m in the *x*-direction, 2.156 m in the *y*-direction, and 1.846 in the *h* direction. The generated DSM meets the requirements of topographic mapping in areas for scale 1:5000 [34]. Therefore, the airborne Ka-band InSAR technology provides a feasible option for large-scale topographic mapping and three-dimensional modeling when it is difficult to carry out other observational techniques.

## 6. Conclusions and Discussion

In this paper, we introduce a topographic modeling experiment for DSMs and DOMs with airborne Ka-band fixed-baseline InSAR in flat and mountainous topographies of China. The experimental data were acquired and processed, and the results analysis and accuracy assessment were finished for evaluation. The key conclusions of this paper are as follows:

1. The work in this paper concerns validation experiments of the airborne Ka-band InSAR system for large-scale topographic mapping and three-dimensional modeling. The airborne Ka-band InSAR has high spatial resolution and high coherence, in addition to the characteristics in full time and all weathers. The DOMs and DSMs from the experiments provide detailed and precise descriptions of topographic features.
2. The whole data processing flow of airborne InSAR data is designed and implemented, including interference processing within SLC scenes, PU, global adjustment of interferometric parameters based on sparse GCPs and TPs between SLC scenes, and generation of a DSM and DOM based on the calibrated parameters. The parallel processing methods based on GPUs are applied for higher processing efficiency. The proposed data processing scheme works well in the experiments.

3. The experimental areas were selected in different topographies in China, including flat (Heyang) and mountainous (Shibing and Qionglai) areas. The airborne InSAR system can complete data acquisition in the topographies, and the generated DOMs and DSMs are qualified. That verifies the high feasibility of the airborne Ka-band InSAR system for topographic mapping and three-dimensional modeling in different topographies.

4. The error indexes of obtained DSMs and DOMs in the experiments meet the accuracy requirements for scale 1:5000 in typical topographies according to the result analysis. It proves the feasibility of topographic mapping and modeling with the airborne Ka-band InSAR system for large-scale DSM and DOM projects.

Although the experimental results verify that the accuracy of airborne Ka-band fixed-baseline InSAR for topographic modeling meet the criterion, there are still some issues that need to be noted for DSM and DOM production generation.

1. The width of the SAR strips from the airborne InSAR system is near 3 km, which may be suitable for modeling in a single frame. However, large-scale topographic mapping tasks, which involve larger areas, will require more strip splicing, and the need for accuracy control would be more prominent. We need to find a resolution for strip processing in large areas.

2. Due to the shadow and layover limitations in mountainous areas, a proper interpolation method or antiparallel flight to fill in non-data areas is required.

3. To remove the corresponding gross error and obtain the corresponding DEM of the experimental area, we need an appropriate filtering algorithm to filter the generated DSM.

**Author Contributions:** Conceptualization, Z.S., H.G. and L.W.; methodology, Z.S., H.G. and Q.X.; software, Y.L.; validation, Y.L., J.G. and L.W.; formal analysis, J.G.; investigation, L.W. and Y.L.; resources, Z.S.; data curation, Y.L.; writing—original draft preparation, Y.L., Z.S. and J.G.; writing—review and editing, J.G., Z.S. and Q.X.; visualization, Y.L. and J.G.; supervision, H.G. and L.W.; project administration, Z.S.; funding acquisition, Z.S. All authors have read and agreed to the published version of the manuscript.

**Funding:** This research was funded by the Key R&D Program Projects in Hainan Province (No. ZDYF2019008) and the Strategic Priority Research Program of the Chinese Academy of Sciences (Grant No. XDA19030104).

**Institutional Review Board Statement:** Not applicable.

**Informed Consent Statement:** Not applicable.

**Data Availability Statement:** The data underlying this study are available on request.

**Acknowledgments:** The authors would like to thank the anonymous reviewers and the editor for their constructive comments and suggestions concerning this paper.

**Conflicts of Interest:** The authors declare no conflict of interest.

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
