# Peer review of "Experimental Results of Three-Dimensional Modeling and Mapping with Airborne Ka-Band Fixed-Baseline InSAR in Typical Topographies of China"

_remotesensing, doi:10.3390/rs14061355_

Round 1

Reviewer 1 Report

This is a very complete paper describing the signal processing steps to generate DSM and DOM from a pair of SLCs from a Ka-band single pass interferometer.  The authors provided a very detailed and methodical description of the processing flow, organizing the steps into three major blocks: interferometric processing, strip calibration, and model building.  The authors then went into more details on 3 critical steps: phase unwrapping in InSAR processing, block adjustment for strip calibration, and model building with three-dimensional point clouds.  The results at the three test sites look very good.

The major issue with this methodology is that GCPs (corner reflectors in this case) with known 3-D coordinates are needed to conduct strip calibration to adjust errors in baseline length, baseline angle, and phase offset.  Given each strip is about 1 km wide, it seems this approach is not very practical for large scale production.  The authors may wish to address this point in the conclusion.

Specific comments:

Table 1 (page 4) – suggest adding the chirp bandwidth and antenna baseline

Line 248 – suggest changing analyzing to analysis

Figure 6 on page 11 – suggest adding unit to the color bar (radians?)

Figure 7 on Page 12 – suggest adding unit to the color bar

Figure 8 on page 12 – what are the units on the sensitivity plots?

Line 322 – why are roll, pitch, yaw set to those values?  Are those from the specific flight?

Line 332 – delete the word “carrier”

Line 348 – Figure 9 label: (blue lines) should be (green lines)

Accuracy assessment of DSMs (Table 3 on page 18)

  RMSE in Y direction in Shibing is a factor of 2 better than the other two sites.  Why?

What’s the expected height accuracy based on instrument design?  Are you assessed accuracies consistent with the expected accuracies?  What are the potential error sources?

Reviewer 2 Report

This paper deals with airborne topographic mapping using coherent millimeter-wave single-pass electromagnetic energy. 

The article is an interesting application. 

The thing that is missing is the following: you have to compare some topographic line with other lines acquired by other sensors, I thought this:
SRTM, DEM generated by Sentinel, and only if you have it, other DEM acquired by LiDAR (but only if it is available). 

Then, calculate me the spectrum of the DEMs and compare them. I expect that your DEM (as it is more detailed than SRTM and Sentinel), has a higher occupation bandwidth.

Finally you have to consider the disturbance generated by the aircraft you use, (if any), and how it affects your DEM. Such perturbation I think is there, even if the interferometric system you use is rigidly installed on the observation platform.

Reviewer 3 Report

The manuscript reports experimental results, assessments and processing procedures of an airborne Ka-band fixed-baseline InSAR over three areas in China. It includes all the stages from data acquisition, handling, processing, presentation and analysis of the experiment with the system. It would be interesting for the readers who are planning and doing research using such systems for three dimensional topographic mapping with SAR systems.

Reviewer 4 Report

In effect, it is a good example of DEM production by Airborne InSAR, but all that is (properly) described in the article is already a standard technique since many years. The only possible novelty is "in typical topographies of Chine", but China is not so different from the rest of the world about topography! 

Round 2

Reviewer 2 Report

Accepted

Reviewer 4 Report

The authors did not reply to may concern: this work does not present new results with respect to state-of-art